# Evaluating Spatial Scenarios for Sustainable Development in Quito, Ecuador

**Esthela Salazar** [1,2,*] **, Cristián Henríquez** [1,3] **, Richard Sliuzas** [4] **and Jorge Qüense** [1]

1   Instituto de Geografía, Pontificia Universidad Católica de Chile, Macul 4860, Santiago, Chile; cghenriq@uc.cl (C.H.); jquense@uc.cl (J.Q.)
2   Departamento de Ciencias de la Tierra y la Construcción, Universidad de las Fuerzas Armadas-ESPE, Sangolquí 171103, Ecuador
3   Center for Sustainable Urban Development–CEDEUS, El Comendador 1916, Providencia, Santiago, Chile
4   Faculty of Geoinformation Science and Earth Observation, University of Twente, 7514 AE Enschede, The Netherlands; r.sliuzas@utwente.nl
*   Correspondence: esalazar1@uc.cl

**Abstract:** Peripheral urban sprawl configures new, extensive conurbations that transcend current administrative boundaries. Land use planning, supported by the analysis of future scenarios, is a guide to achieve sustainability in large metropolitan areas. To understand how urban sprawl is consuming natural and agricultural land, this paper analyzes land use changes in the metropolis of Quito, considering a combination of urban planning, natural conservation and risk areas. Using the Dyna-CLUE model we simulate spatial demands for future land uses by 2050, based on two growth scenarios: the trend scenario (unrestricted growth) and the regulated scenario, which considers two parameters—a government proposal for urban expansion areas and laws that protect natural areas. Both scenarios show how urban expansion consumes agricultural and natural areas. This expansion is backed by urban policies which do not sufficiently account for conservation areas nor for risk areas. Therefore, these simulations suggest that planning should follow a more holistic approach that explicitly considers urban growth beyond current administrative limits, in what we refer to as the New Metropolitan Area of Quito.

**Keywords:** urban growth; Dyna CLUE; land use modeling; urban planning

## 1. Introduction

Urban sprawl is a phenomenon that is changing landscapes throughout Latin America [1]. The physical expansion of cities occurs at the expense of agricultural, forest or natural areas without taking into account whether these lands were designated for urban land use or not, thereby deteriorating the natural resources that sustain the city [2]. Increasing land consumption and relatively low population density have become features of the peri-urban areas of major cities [3]. The transition from monocentric radial cities to cities that expand physically, economically and functionally is evident in the main urban areas of Latin America. They transform, forming both continuous and discontinuous morphologies that structure a new territorial and planning scale [4].

This dynamic of urban growth implies changes in land use that respond to socio-economic, environmental and physical factors that promote urbanization [5]. Land use planning attempts to reduce the negative impacts of such growth. It tries to influence the dynamics of land use changes so as to achieve configurations that balance the needs of all actors in a specific territory [6]. Urban planners and decision-makers require accurate and detailed information on the urban growth potential, land use change processes, locations and spatial patterns to support their guidance functions, to evaluate and plan for future development and to prevent territorial conflicts [7].

Simulated scenarios of future land use configurations allow us to know the impact of spatial policies on land use transformations. "Scenarios are fictional, they can serve as artificial case histories which illustrate the implications of policies which might be ignored if examples only from the past real world are considered" [8]. Thus, simulation models based on possible scenarios that combine demands, regulations and the views of the main stakeholders allow the future to be analyzed in a more comprehensive and structured way and have become a fundamental planning tool [9]. Currently, models are most widely developed and implemented in the United States, Asia, China and Europe. There is little application of these models in Latin America [4,9], especially for urban analysis. Most such models have focused on agricultural or forestry uses [4,10].

There are several benefits of using scenario modeling for decision making. First, scenarios provide heuristic support to explain events and their consequences [11]. Facts and data, in themselves, do not make much sense until they are related within a framework that includes the interaction of social, economic, political and technological factors. Scenarios are an appropriate means to relate and understand isolated pieces of information within the same framework [11]. Second, the description of a scenario requires a certain level of specificity (the who, what, where, when and why of an action). This enables decision-makers to consider factors that could be overlooked if only abstract principles and general statements were to be considered. Third, being fictitious or artificial constructions of future events and situations, scenarios illustrate and inform the political implications that could be ignored if only real-world examples or past evidence were considered. In short, scenarios provide a means to show and explore options to facilitate the discussion among planners, stakeholders, different professionals, disciplines and levels of management regarding future development options.

In this research, a simulation model is applied to the metropolitan area of Quito, Ecuador, to explain and analyze the dynamics of urban growth by 2050 based on two scenarios: trend and regulated. Both scenarios were validated based on relevant planning instruments and on the visions of main actors from local government agencies. The formation of the Quito metropolitan area is analyzed as a product of urban growth. Quito's location in a narrow mountain valley at the foot of the active Pichincha volcano results in rather linear morphology that is more than 30 km long and from 5 to 8 km wide [12]. Moreover, much growth occurs outside the administrative limits of Quito City, resulting in a conurbation incorporating the adjacent municipalities and increasing pressure on their natural and agricultural resources.

The ultimate purpose of this research is to answer the following question: Given the simulation of Quito's urban dynamics, is it necessary to create a new metropolitan area for Quito? The simulation model effectively represents the magnitude and locations of possible land use changes and their spatial patterns [9] based upon the opinions of local actors. They can more accurately evaluate and project the possible consequences of urban growth effects, and they can also plan for the sustainable development of Quito and its metropolitan area.

## 2. Materials and Methods

### 2.1. Study Area: The New Metropolitan Area of Quito (NAMQ), Why and What for?

The Metropolitan District of Quito (DMQ in Spanish), or Quito canton, is in the Province of Pichincha in the central highlands of Ecuador, within the hydrographic subbasin of the Guayllabamba River. The DMQ is the political–administrative–economic capital of Ecuador and the most populous city with 2,239,191 inhabitants [13,14] within an area of 423 km$^2$. Its urban population is 1,619,146 inhabitants (72%) and the rural population is 620,045 inhabitants (28%) [13]. It is economically and functionally integrated with the other cantons of Pichincha province (Figure 1). The altitude ranges from 500 to 4780 meters above sea level forming a landscape which is characterized by numerous valleys, mountains and active volcanoes, such as the "Pichincha" volcano, and inactive volcanos such as Ilaló volcano. There are diverse climatic conditions: humid tropical, semiarid and hyperhumid [15]

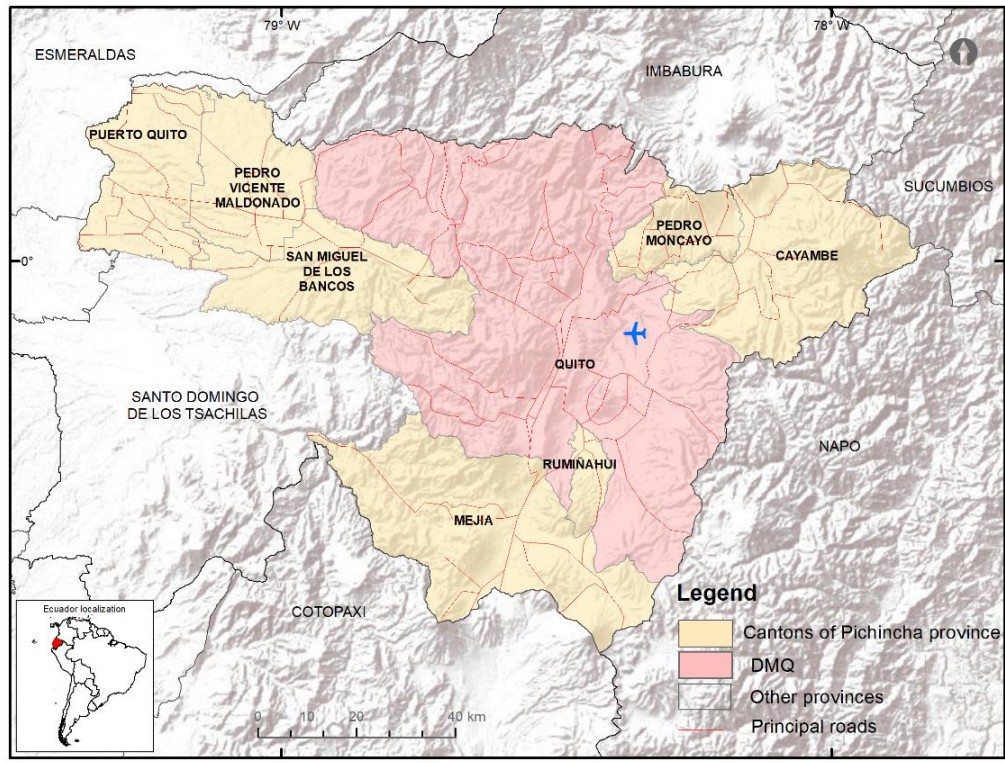

**Figure 1.** Spatial location of the Metropolitan District of Quito (DMQ) and its neighboring cantons in the Province of Pichincha.

The territory of DMQ is divided into 8 Zonal Administrations for the purpose of decentralizing public bodies and improving public service delivery. These areas are further divided into 32 urban parishes and 33 rural parishes (Figure 2).

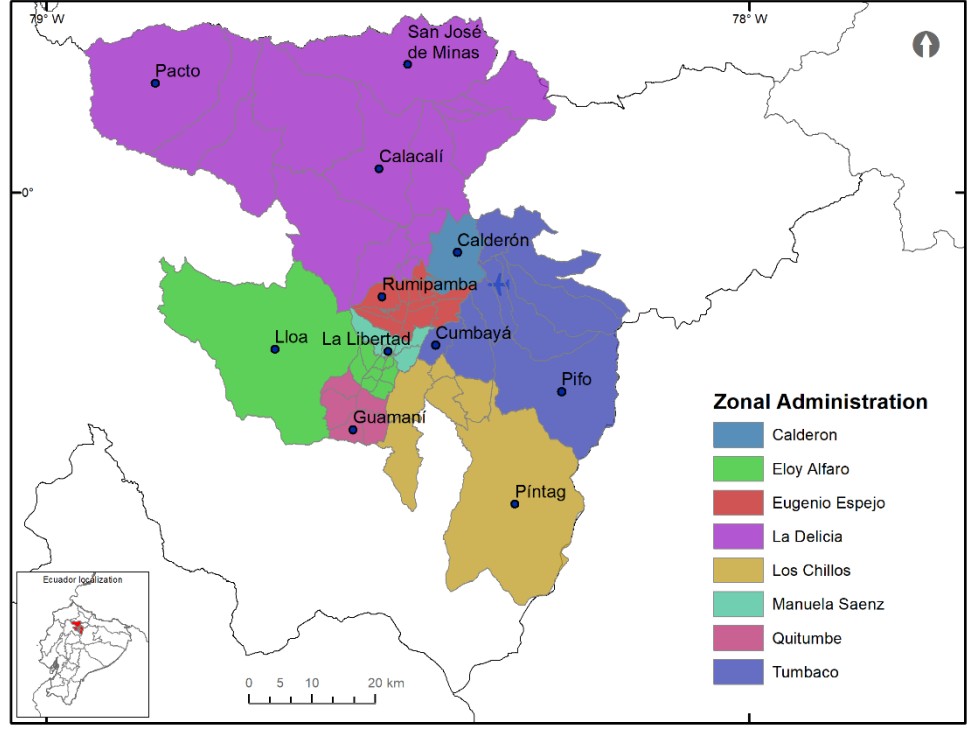

**Figure 2.** Administrative–political division of the DMQ.

Quito is one of the Latin American cities that comparatively recently (1990) joined the group of urban metropolitan conglomerates [16]. This paper describes how land use and land occupation have rapidly transformed over a period of 19 years (1998–2017), exhibiting a clear trend of urban growth in the valleys of Cumbayá, Tumbaco and Los Chillos as the most common growth areas. Additionally, new fronts of growth are emerging towards the neighboring cantons of Pedro Moncayo, Cayambe, Rumiñahui and Mejía.

These cantons have different connections and functional relationships with the DMQ, as follows: the relationship with Mejía canton is commercial due to milk production (approximately 860,000 liters of milk are produced daily for the city of Quito) [17]; for its part, the rural parish Cutuglahua in Mejía, which is adjacent to the urban area of Quito, stands out for its high urban growth rate (16.4%). The cantons Pedro Moncayo and Cayambe have a commercial and labor relationship with the DMQ due to the presence of flower farming, which has become one of the main economic activities of these cantons and of the country (around 90 tons of roses per day reach the Quito international airport from these cantons) as one of the country's primary export products [18].

In the case of Rumiñahui canton, the main activities that connect it with Quito are commerce and education. According to information from the Development and Territorial Planning Plan of Rumiñahui canton 2012–2025, the main immigrant population of Rumiñahui comes from Quito [19]. A common denominator in these cantons that allows the strengthening of relations is the Panamericana Highway, which connects Quito with neighboring cantons and with other provinces such as Ibarra, Tulcán, southern Colombia and the central Sierra of Ecuador (Figure 1).

The expansion of the urban area of Quito has already exceeded its administrative boundaries. This work proposes fusing a single conglomerate, which we refer to as the New Metropolitan Area of Quito (NAMQ), to be formed by the five cantons with close physical, economic and social relations: Quito, Pedro Moncayo, Cayambe, Rumiñahui and Mejía. For this paper, these cantons together form a new functional unit for the analysis of land use changes and the simulation of future territorial planning scenarios.

The analysis considers several important instruments for the planning and design of the territory and the organization of urban growth in order to achieve a harmonious and ecologically sustainable development (Table 1).

**Table 1.** Urban planning instruments.

| | | |
|---|---|---|
| Legal instruments at the national level that allow territorial planning to proceed | The Code of Territorial Organization, Autonomies and Decentralization (COOTAD) | It is responsible for organizing the powers of the different levels of government (regional, provincial, cantonal and parochial, especially of the decentralized autonomous governments (GADs)). Territorial planning stipulates that it must be complementary to the GADs of its constituency [20,21]. This applies in the case of Quito and the neighboring areas, so that territorial planning and administration among municipalities can be coordinated jointly. In the case of Quito, the Metropolitan Land Use Plan (PMOT) and its instrument, the Land Use and Occupation Plan (PUOS) are determined, establishing land use, compatibility relations, occupation, buildability, authorization, road categories and dimensions and areas of involvement and special protection. |
| | The Law of Territorial Planning, Use and Management of the Land (LOOTUGS) | It establishes the principles and rules that govern the territorial planning, use and management of urban and rural land. Regarding land management, the development plans and territorial planning of municipal and metropolitan (GADs) areas must contain a land use and management plan [21]. In the case of Quito and the neighboring cantons, this law grants the power to expropriate, reserve and control areas for future urban development. |
| | The Law of Rural Lands and Ancestral Territories | This law is intended to regulate the use and access to ownership of rural land. The right to ownership of it that must fulfill the social and the environmental function [22] determines the expansion of urban areas in rural lands without agricultural suitability under the authorization of the National Agrarian Authority. In the case of Quito and the neighboring cantons, this law regulates and protects rural lands where fragile ecosystems exist or belong to the national system of protected natural areas (SNAP). |

Source: Authors' elaboration.

In addition, we consider other information relevant for analysis and discussion with the scenarios, such as agricultural crops, natural protected areas (SNAP), forests declared by the Ministry of Environment and volcanic risk areas declared by the Geophysical Institute (IGENP) and the National Risk and Emergency Management Service (SNGRE). However, it must be borne in mind that these areas are only considered indicatively. They do not entail binding restrictions for urban planning, as will be shown in Sections 3 and 4.

### 2.2. Land Use Modlling: Dyna-CLUE

This research used the "The Conversion of Land Use and its Effects modeling framework" (Dyna-CLUE) model. An advantage of this model is the use of several nonspatial driving factors such as population growth, urban growth patterns and demands of exogenous elements, in addition to the physical and socioeconomic factors that determine the pressure on land use change [8]. Verburg [23] defines land use change models as "tools to support the analysis of the causes and consequences of land use dynamics combining human and natural process". Dyna-CLUE is based on a probabilistic logistic regression method [23]. It is an explicit model which generates a multilayer spatial and quantitative description of land use changes, including the simulation of likely future land use changes. The Dyna-CLUE methodology analyzes land use systems as complex multi-level systems at the interface of multiple social and ecological systems (Figure 3). Thus, interactions which are more consistent with the reality of the study area are generated [24].

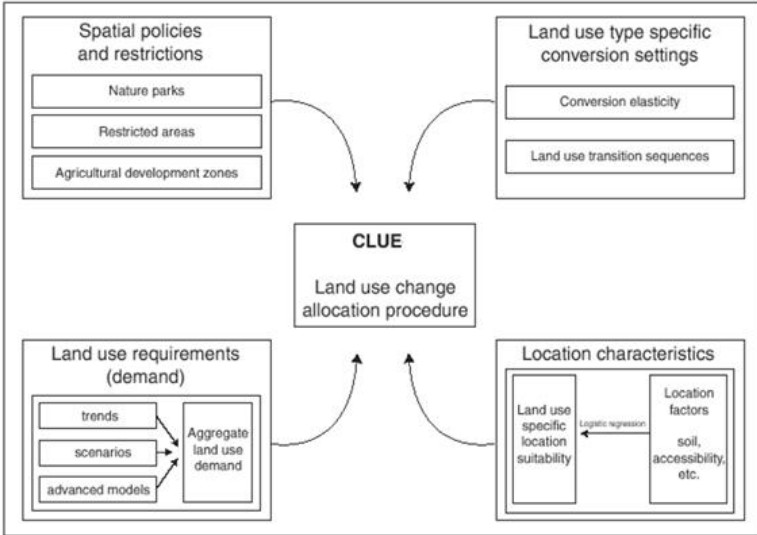

**Figure 3.** General view of the Dyna-CLUE model. Source: The CLUE model Hands-on Exercises Course Material (Verbug, 2010).

We simulated future scenarios for the NAMQ using Dyna-CLUE and evaluated the quantitative outputs. This modeling allowed monitoring the land cover changes in the past and analyzing possible changes in the future [25]. Moreover, the consultation of key actors (decision-makers) guided the modeling process according to the regulations and the instruments for land use planning in the municipality of Quito and the provincial government, as additional inputs taking into consideration their different demands [11]. The nonspatial data were considered in the location of the different types of land demands and the spatial data for the land use allocation procedure. The approach " . . . combines a top-down location of the land use change together with a bottom-up determination, which corresponds to the transition from a specific land use to a different one." [8]. Both are combined in an

interative function that assigns land uses based on the total probability (TPROP$_{i,u}$) of every land use type $u$ for each grid cell $i$, thus:

$$TPROP_{i,u} = P_{i,u} + ELAS_u + ITER_u \tag{1}$$

where $P_{i,u}$ is the suitability of location of every land use type $u$ in grid cell $i$, $ELAS_u$ is land use elasticity $u$, and $ITER_u$ is the iteration variable indicating the relative competitive strength of land use $u$. $P_{i,u}$ is represented as a probability, ranging from 0 to 1, as a result of logistic models; $ELAS_u$ also varies from 0 to 1 and characterizes the ease or difficulty of land use conversion due to bottom-up constraints imposed by society. Finally $ITER_u$ is set by the model and depends on the annual land use demands set by top-down constraints.

### 2.2.1. Land Use Classification and Conversion Setting

For the land use classification maps, two satellite images from 1998 (Landsat 5) and 2017 (Landsat 8) were used. These images, with a resolution of 30 m, were downloaded from the United States Geological Survey (USGS) [26]. With the 2017 image, we worked on a subscene adjusted to the boundaries of the cantons and parishes adjacent to the DMQ to determine the NAMQ. A supervised classification (based on the spectral signature) in ArcGIS was conducted based on training sites (polygons) according to the Corine Land Cover (CLC) methodology initiated by the Coordination of Information on the Environment (CORINE) [27] program: (1) urban; (2) moorland; (3) native vegetation; (4) without vegetation; (5) scrub; (6) agricultural; (7) water bodies; and (8) without data (clouds or ice). A majority filter was applied to each classifed output to merge isolated pixels with the class with the largest number of neighboring pixels. Then, a group of regions was created as a second filter to eliminate regions of less than 10 pixels and assign them as no-data. The non-data areas were the reassigned to the majority class within their neighborhood (Figure 4). The main transitions of land use between 1998 and 2017 were analyzed to determine which conversions should be considered in the model (Tables 2 and 3). For modeling purposes, the eight land use classes were reduced to four classes which were considered most useful to describe the major land use dynamics: (1) urban; (2) native vegetation; (3) scrub; (4) agricultural. In this process, all small land use areas (<10 pixels) were merged to the majority class within their neighborhood.

**Table 2.** Land use and land cover in the New Metropolitan Area of Quito (NAMQ), 1998 and 2017.

| Land Use and Cover | ha 1998 | % | ha 2017 | % | Difference |
|---|---|---|---|---|---|
| Urban | 23,880 | 4% | 71,516 | 13% | 8% |
| Agricultural | 201,672 | 36% | 117,642 | 21% | −15% |
| Native vegetation | 107,888 | 19% | 178,136 | 32% | 12% |
| Moorland | 127,626 | 23% | 138,010 | 24% | 2% |
| Scrub | 75,364 | 13% | 48,122 | 9% | −5% |
| Without vegetation | 2914 | 1% | 1187 | 0% | 0% |
| Bodies of water | 253 | 0% | 151 | 0% | 0% |
| Clouds or ice | 25,711 | 5% | 10,545 | 2% | −3% |
| Total | 565,308 | 100% | 565,309 | 100% | 0% |

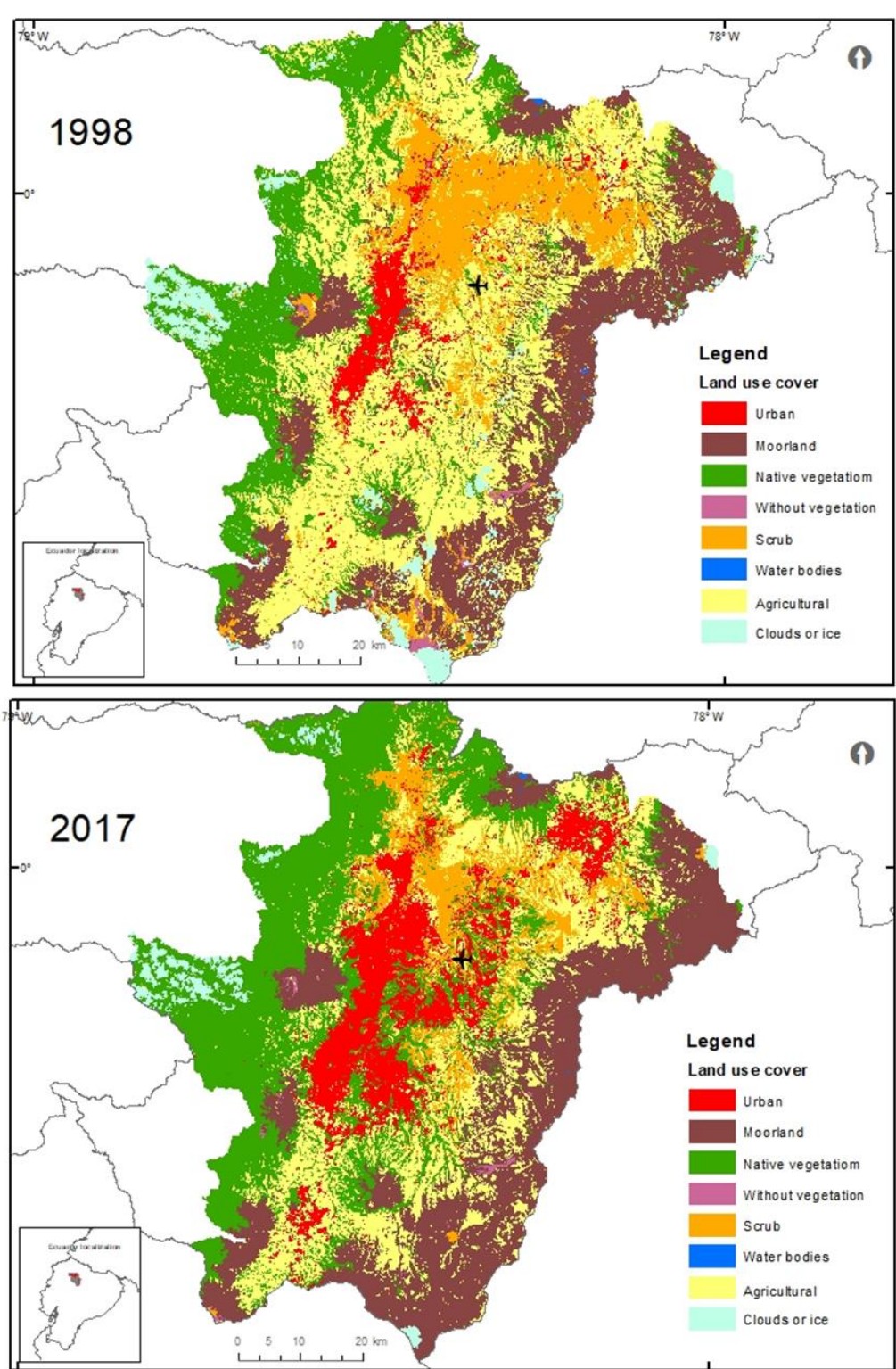

**Figure 4.** Land use and land cover: 1998 in contrast to 2017.

**Table 3.** Transitions to urban and agricultural land use from 1998–2017.

| Transitions | Change as a Perecntage of Total Area | % Change per Transition Type |
|---|---|---|
| Agricultural to urban | 2.84 | 0.47 |
| Scrub to urban | 1.38 | 0.23 |
| Native veg. to urban | 0.84 | 0.14 |
| Scrub to agricultural | 0.98 | 0.16 |

### 2.2.2. Driving Forces

With the support of main government agencies and literature, nine main driving forces for urban growth were identified (Table 4 and Figure 5). These variables can potentially affect the specific location of land uses. For example, the main road network is a primary factor for population growth due to accessibility to the territory. Educational infrastructure and health establishments are also considered to attract population growth. Moreover, aqueducts allow irrigation of agricultural areas whereas the slope affects both the location of agriculture and human settlements.

**Table 4.** Driving Forces.

| N. | Driving Forces | Unit of Measure | Source |
|---|---|---|---|
| 1 | Aqueduct | Distance (m) | Geographic military |
| 2 | Hypercenter | Distance (m) | MDMQ |
| 3 | Towns | Distance (m) | Geographic military |
| 4 | Health establishments | Distance (m) | Ministry of Health |
| 5 | Train line | Distance (m) | Geographic military |
| 6 | Main road | Distance (m) | Planning secretary |
| 7 | Airports | Distance (m) | Geographic military |
| 8 | Educational establishments | Distance (m) | Ministry of education |
| 9 | Slope | Degrees | Landsat 2017 |

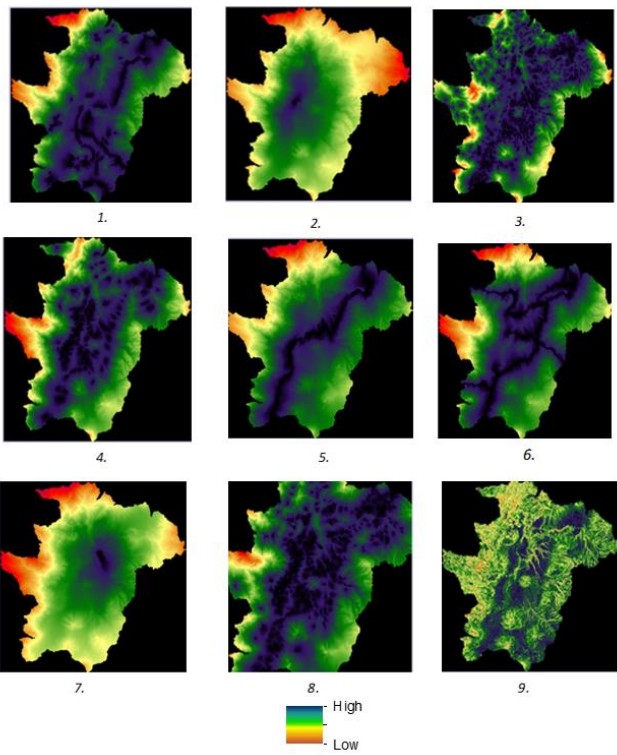

**Figure 5.** Spatial representation of driving forces.

We determined the relationship between the driving forces and land use. First the impedance (geometric distance between two points) was calculated based on the slope. These distances were normalized between values of 0 (closer) and 1 (farther). Second, the correlations between forces and land use were calculated with the logistic regression formula used by Dyna-CLUE. Cramer's V Test [28] was used to show how much the driving force explains a given land use (a value greater than 0.15 is considered acceptable; a value greater than 0.4 is considered very good). Third, the importance of the driving forces was verified using Pearson's coefficient [29]. Fourth, the logistic regression with each of the driving forces was calculated to obtain the necessary coefficients required by Dyna-CLUE. In addition, the receiver operating characteristic (ROC) curve value [30], which serves as a measure of discrimination in the model, was obtained. A ROC value of over 0.7 is considered acceptable (Table 5); the closer to 1 that the model predicts, the more likely an event will occur in a given area.

**Table 5.** Logistic regression results (coefficients) for each type of land use to be modeled. The value of the receiver operating characteristics (ROC) curve is included in the final line as a model descriptor.

| Driving Factors | Urban | Native Vegetation | Scrub | Agriculture |
|---|---:|---:|---:|---:|
| Distance from airport | −2.9852 | | | |
| Distance from educational establisments | −39.0324 | | | |
| Distance from aqueduct | | 4.4887 | −5.0209 | −4.2688 |
| Distance from heatth establishments | −3.1558 | | −5.2008 | |
| Distance towns | −16.6785 | | | |
| Distance train line | | 3.4340 | 4.7521 | |
| Distance roads | −5.4759 | 2.4874 | −6.5965 | −4.8078 |
| Slope | 16.0011 | 5.4492 | 6.2952 | 7.6178 |
| Distance hypercenter | 0.1350 | −0.3494 | | 2.3745 |
| Intercept | −5.1629 | −4.7355 | −4.4391 | −4.8062 |
| Roc | 0.95 | 0.94 | 0.82 | 0.86 |

### 2.2.3. Calibration and Validation of the Model

The model was calibrated using two satellite images from 1998 and 2017. For the calibration, the Cross Tab or contingency matrix proposed by Pontius [31] was used to obtain the values of Row (initial year), Columns (real year) and Plane (modeled year). These values allow distinguishing pixels that are correct due to persistence and pixels that are correct due to change (hits). On the one hand, there are pixels where a change is predicted but a persistence occurs (misses), and on the other hand, there are pixels where the model predicts persistence but a change occurs (false alarm). Two validations of the model were performed. The second, using the following elasticity values per class (1—urban, 0.6—natural vegetation, 0.5—scrubland, 0.7—agricultural), produced the best results (Figure 6). Scrubland has the lowest value (0.5), which means that it is most easily converted to another land use.

For the final evaluation, the figure of merit [32], which shows the percentage of successes, losses and false alarms was used. Pontius [32] reports that the values of the figure of merit are related to the size of the pixel. In this study, the pixel size is $30 \times 30$ m, which explains the high value of the figure of merit. This calculation was performed based on the Excel table provided by Pontius [33].

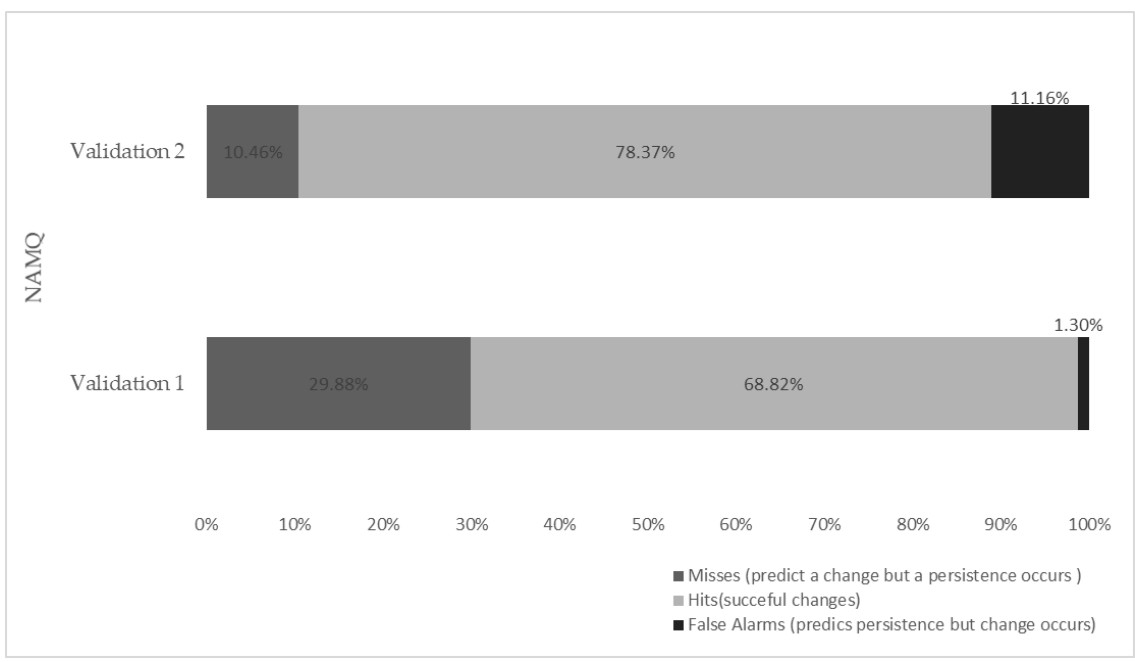

**Figure 6.** Merit figure between observed and predicted 2017 maps.

2.2.4. Land Use Demands

For modeling the land use, categories (urban, native vegetation, scrub and agriculture) were used. The basic land use demands were based on nonspatial and discretionary numerical data [34] that were defined for each land use and for each year. These were attained from historical trends (1998–2017) derived from the satellite images. In addition, estimations based on the National Institute of Statistics and Censuses INEC [35] and the intercensal growth rate for Ecuador (1.9%) were used for urban use and extrapolated to the 2050 modeling year. Consequently, a linear projection based on the linear regression provided the basis to establish the demands for other land uses (native, scrub and agricultural vegetation) and for their extrapolation (Table 6). A constant value for scrub coverage was assumed throughout.

**Table 6.** Estimated land use demands 2017–2050.

| Land Use | Demand (ha) | | | |
|---|---|---|---|---|
| | **2017** | **2027** | **2037** | **2050** |
| Urban | 71,516 | 86,604 | 102,468 | 122,655 |
| Native vegetation | 328,029 | 324,672 | 318,541 | 312,308 |
| Agriculture | 117,642 | 106,910 | 96,177 | 82,224 |
| Scrub* | 48,122 | 48,122 | 48,122 | 48,122 |

* A constant value assumed. Source: INEC-2010; interpretation of satellite images 1998–2017.

2.2.5. Future Scenarios: Trend and Regulated

Two scenarios were developed. Scenario 1, the trend or business-as-usual (BAU) [2] scenario, shows the growth trend in the current context until the year 2050, i.e., growth under conditions with little or no regulation. It is based on the premise that changes in land use (influenced by factors or driving forces such as roads, equipment or land market) that have happened in the past will continue. This scenario entails extended urbanization and the reduction of rurality [3], reaffirming the process of urban fragmentation that transforms the traditional city [36].



Scenario 2, the regulated scenario, considers the areas of urban expansion proposed by the Ecuadorian Space Institute (IEE 2013) (Figure 7), determined by the Law of Territorial Planning, Land Use and Management (LOOTUGS) [21] (Table 1). It should be noted that the Land Use and Occupation Plan (PUOS) for the DMQ was not used because it does not determine the areas of urban expansion nor do the Development and Land Management Plans (PDOTs) of the other cantons under study. The influence of natural protected areas and hazardous areas on this scenario is considered in the discussion section below.

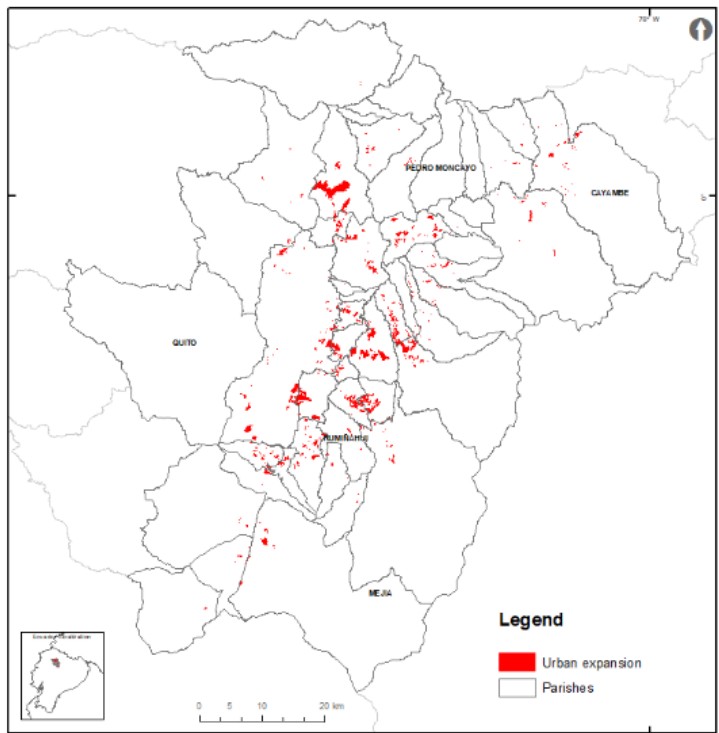

**Figure 7.** Urban expansion areas established by the Ecuadorian Space Institute (IEE), 2013.

## 3. Results

The simulation of future scenarios (2017–2050) allows assessing the potential impact of territorial policies that enable or restrict urban expansion [1] on the current trends of demographic dispersion and land use. The results of the BAU scenario will be followed by the regulated scenario.

### 3.1. Trend Scenario or BAU: Fragmented Urban Expansion

This scenario reveals a tendency toward spatially fragmented urban expansion that extends from the central metropolitan area to form a conurbation including the neighboring cantons of Rumiñahui and Mejía to the south, as well as Cayambe and Pedro Moncayo to the north. Consolidation of new urban conurbation occurs especially close to the main transportation routes (E35, Aloag–Santo Domingo, Calacalí–la Independencia) and main facilities, according to the result of the logistic regressions.

By 2050, an urban growth trend of 23% is estimated in comparison to the year 2017 (Table 7). Urban growth is directed towards two eastern and western flanks of the NAMQ: to the east, towards the parishes located near the new Quito airport and to the northeast, towards the parishes of the cantons Cayambe and Pedro Moncayo. In this scenario, urban growth occurs in all rural parishes attached to the urban center and towards the southeast, forming a conurbation including Rumiñahui canton. To the south, growth is directed towards Mejía canton, mainly in the parishes adjacent to the city of Quito greater metropolitan area and towards Mejía canton. In the west, urban expansion

is fragmented in the direction of the Calacalí–la Independencia road and towards the Aloag–Santo Domingo road (Figure 8).

This simulation estimates that the agricultural area will be reduced from 21% to 17% in 33 years, mainly due to urban expansion, especially in the northeastern part of the metropolitan area, e.g., Cayambe and Pedro Moncayo cantons where flower plantations are primarily located. Native vegetation shows a particular behavior because there was an increment between 1998–2017 due to the implementation of several regulations for conservation in the DMQ and the new declarations of natural protected areas [15]. However, by 2050 the natural vegetation will reduce from 58% to 53% due to urban expansion, mainly on the eastern and western flanks that make up the green belt around the urban area of Quito. The Quito conurbation will then include the valleys of Tumbaco, Cumbayá, Nayón and Conocoto and Rumiñahui canton. Scrubland, on the same eastern and western slopes around the urban area of Quito, is also expected to reduce to 7% due to urban expansion. However, the conservation of small scrub remnants is observed in the northeastern parishes of the DMQ (Guayllabamba, San Antonio) and in Pedro Moncayo canton (Figure 8).

**Table 7.** Land use/cover of business-as-usual (BAU) and regulated scenarios by 2050.

| Land Use | ha 1998 | % | ha 2017 | % | ha Trend Scenario 2050 | % | ha Regulated Scenario 2050 | % |
|---|---|---|---|---|---|---|---|---|
| Urban | 23,880 | 4% | 71,516 | 13% | 131,220 | 23% | 140,749 | 25% |
| Nat.Veget. | 264,391 | 47% | 328,029 | 58% | 298,456 | 53% | 337,503 | 60% |
| Scrub | 75,364 | 13% | 48,122 | 9% | 40,787 | 7% | 551 | 0% |
| Agriculture | 201,673 | 36% | 117,642 | 21% | 94,845 | 17% | 86,506 | 15% |
| Total | 565,309 | 100% | 565,309 | 100% | 565,309 | 100% | 565,309 | 100% |

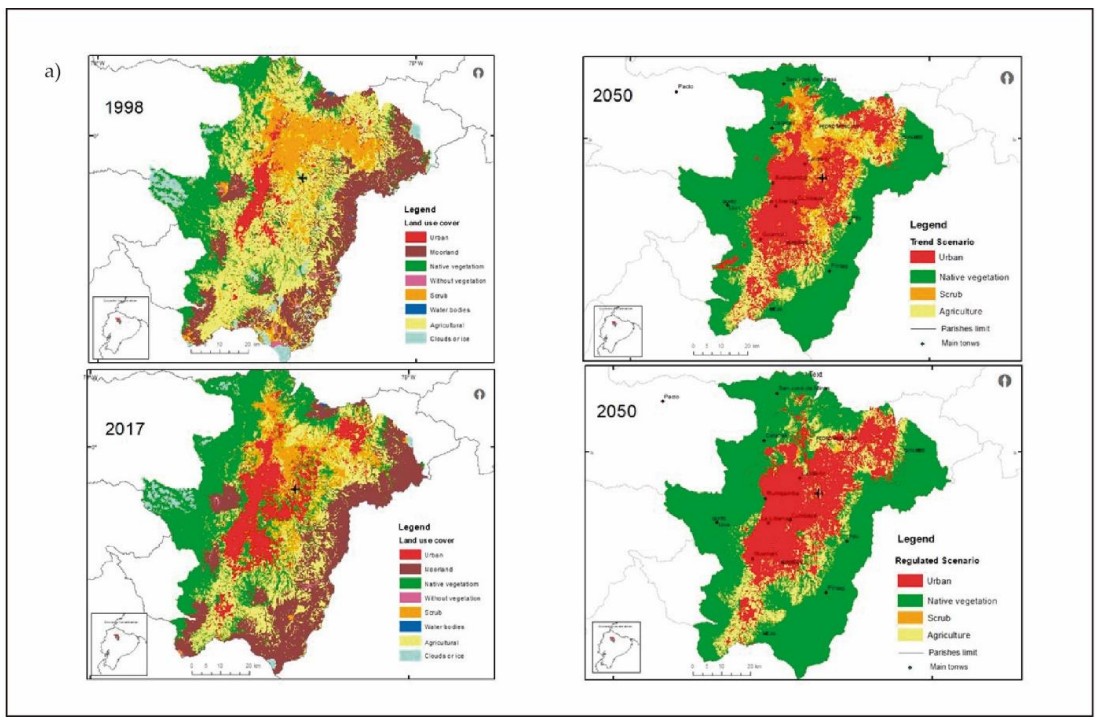

**Figure 8.** *Cont.*

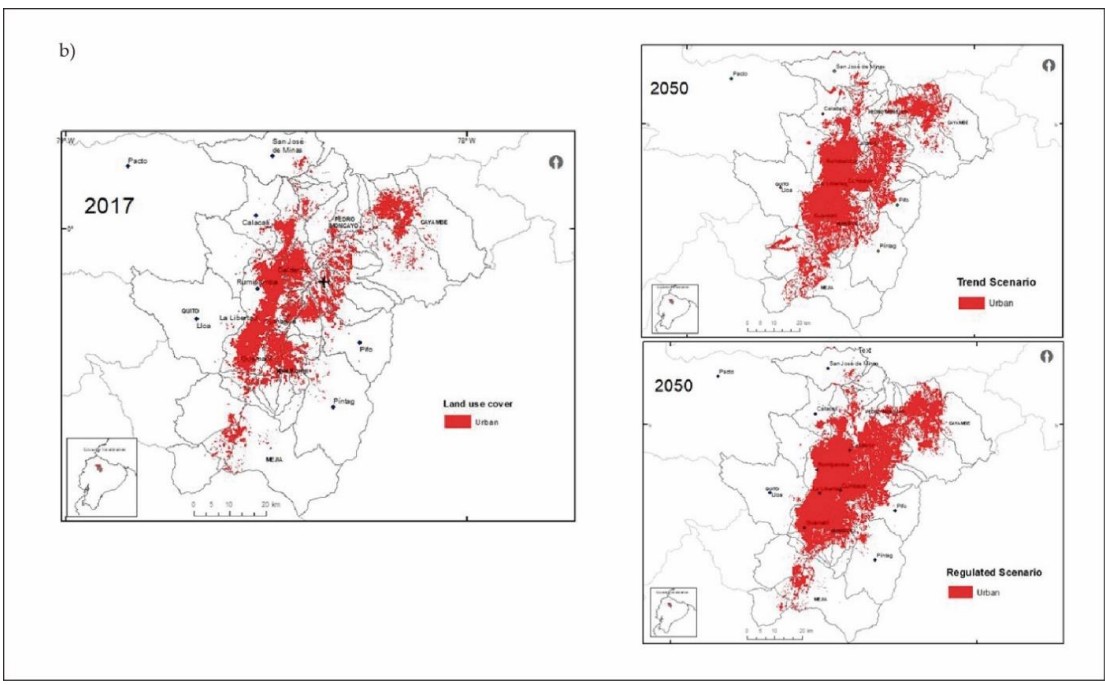

**Figure 8.** (**a**) Land use and cover 1998–2017 and 2050 (**b**) Urban expansion 2017 and regulated scenario vs. trend scenario 2050.

### 3.2. Regulated Scenario: Land Use Planning

This scenario simulates and implements urban growth according to the IEE's urban expansion proposals, which set two criteria for qualifying areas such as urban expansion based on orthophotos or satellite images [37]: prevalence of buildings or constructions on agricultural use; and the prevalence of road layouts in the area. However, this information is not used for the territorial planning because the municipality and the provincial government have not validated this information.

Interestingly, urban expansion is 2% higher in the regulated scenario (25%) compared to the trend scenario (23%) (Figure 9). The expansion areas are located around the already consolidated urban areas. In addition, to the northeast of the NAMQ, there is a large urban concentration forming an urban corridor with the cantons Pedro Moncayo and Cayambe. Unlike the trend scenario, urban growth to the south is more restricted, respecting the remnants of natural vegetation (around the Ilaló volcano) and agricultural areas in Mejía canton. However, the total loss of agricultural area is 2% higher (15%), particularly on the eastern and northeastern flanks in Cayambe canton (Table 7). We analyzed how in both scenarios, urban sprawl will replace agricultural areas (flowers, fruits and cereals), mainly in Cayambe and Pedro Moncayo cantons, (Figure 9) showing the regulated scenario and agricultural crops.

In the regulated scenario, scrubland disappears almost completely in the northern area of the DMQ (Guayllabamba, San Antonio, Malchingui) in Pedro Moncayo canton (Tocachi) and along the eastern flank from Cayambe to Rumiñahui, leaving only a small remnant in San Antonio and Tocachi that does not reach a representative percentage (Table 7). In the regulated scenario, the higher protection of natural areas in the regulated scenario occurs at the expense of scrubland.

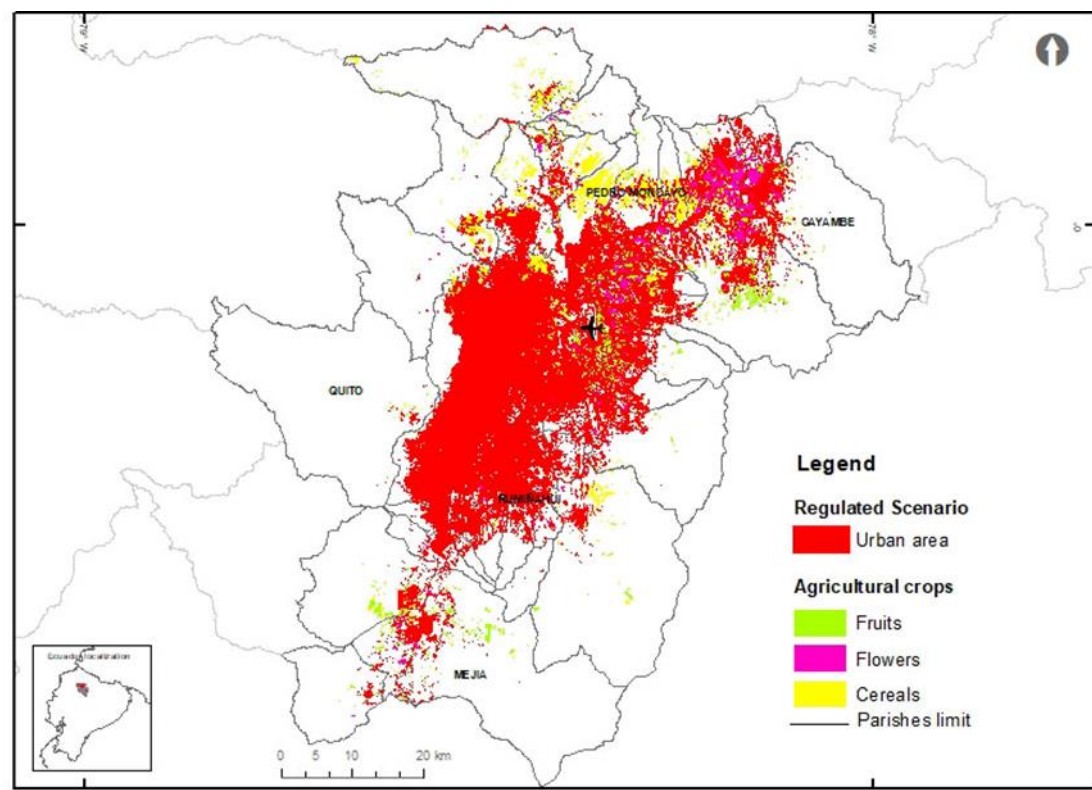

**Figure 9.** Regulated scenario (urban areas in 2050) and agricultural crops. Source: Ecuadorian Space Institute, 2013.

## 4. Discussion

The results from the trend and regulated scenarios reinforce the earlier findings on urban growth trends in Quito [38,39]. From the central metropolitan area of Quito, growth is projected towards the periphery, forming a conurbation with the neighboring cantons Pedro Moncayo and Cayambe (to the north) and Rumiñahui and Mejía (to the south). This particular case is a clear example of the process of metropolitan formation in Latin American [40].

In the trend scenario, urban growth (23%) is lower than in the regulated scenario (25%), which is very unexpected. In the former, growth is directed normal towards the main road transportation routes and the zones with better services and facilities. It is a somewhat fragmented and dispersed growth, by way of urban patches that increasingly occupy more extensive areas beyond Quito's administrative boundaries. The generation of urban corridors supported by a network of roads that facilitate mobility in search of goods and services in other territories is apparent.

By contrast, in the regulated scenario, the IEE's proposals for urban expansion configure a large urban conurbation that expands eastwards (to where the new Quito airport is located), as well as to the northeast towards Pedro Moncayo and Cayambe. It also expands to Rumiñahui and to a lesser extent to Mejía. Therefore, urban expansion from the city to the periphery is subject to real estate speculation [1] that attracts infrastructure, services and new road infrastructure (connecting roads with the new Quito airport), all acting as driving forces for urban growth [24] analyzed in this paper.

This case confirms how urban growth replaces agricultural land uses, as in other Latin American cities such as Santiago [4]. In both scenarios, urban expansion entails significant loss of agricultural areas (Figure 9). This is important as the horticultural, fruit and milk production in Pedro Moncayo, Cayambe and Mejía cantons are main sources of food for the metropolis [41] and one of the main economic activities of the country [18]. However, the consequences of such conflicting land use transitions may not be fully appreciated and considered in spatial and other policies [42].

In the regulated scenario, the use of two indicative layers corresponding to protected and hazardous areas (see Section 2.1 for details) affects the expansion patterns. As a result, the regulated urban growth consumes the forest area corresponding to the eastern flank and green belt of Quito. To the north, the growth is limited by the presence of a national protected area (Pululahua), but it is also directed towards the northeast of the NAMQ where there is no significative presence of natural protected areas, except for the small Jerusalem Protected Forest in Pedro Moncayo canton. Unlike the trend scenario, regulated urban growth is more compact and it consumes less of the native vegetation, especially the forests (Figure 10).

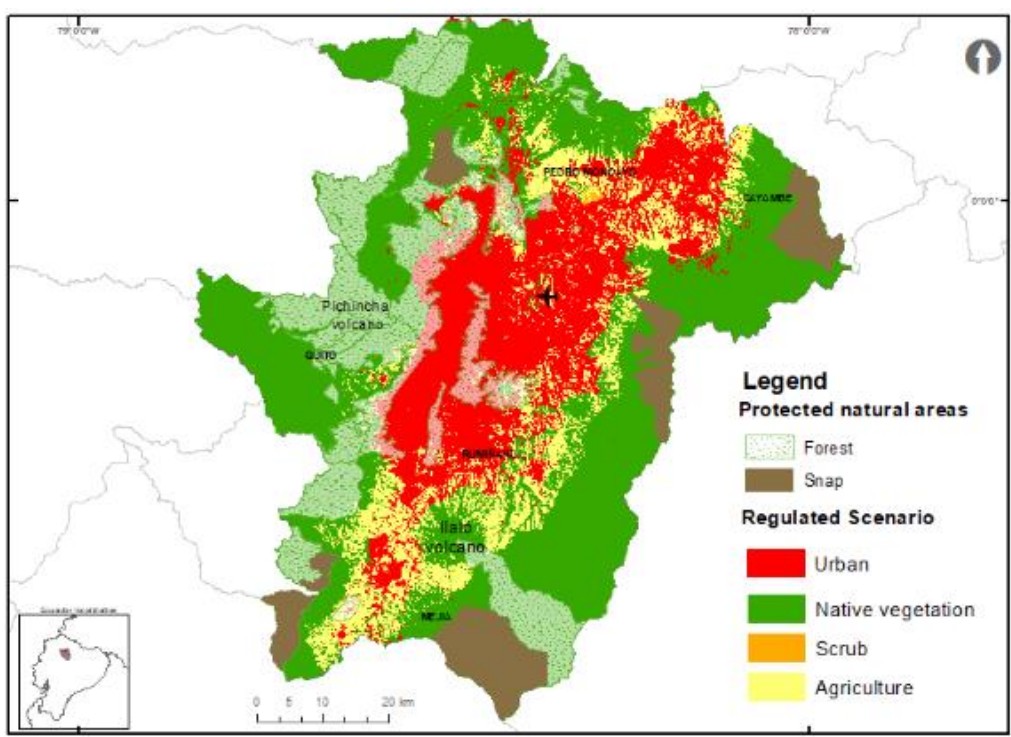

**Figure 10.** Regulated scenario and natural protected areas, 2050. Source: Ministry of Enviroment, 2017.

We also analyzed where regulated urban growth occurs and how it is directed towards the high-volcanic-risk areas declared by the Geophysical Institute (IGENP) and the National Risk and Emergency Management Service (SNGRE) on the western flanks towards the volcanos Pululahua, Guagua Pichincha, Ninahuilaca or Atacazo and on the eastern flank towards the Cayambe and Cotopaxi volcanos. A study published by the French Institute of Andean Studies (IFEA) corroborates how devastating a possible eruption of the Cotopaxi volcano would be for the southeast valleys (Valle de los Chillos) of Quito and parishes of Rumiñahui (Sangolqui, Pintag), especially due to the passage of lahars [43]. Both urban zones and agricultural zones are in high risk areas. For example, the flower plantations in Cayambe canton are in a risk area due to the Cayambe volcano, while cultivation of cereals and flowers in Pedro Moncayo is exposed to a high risk due to the Pululahua volcano (Figure 11).

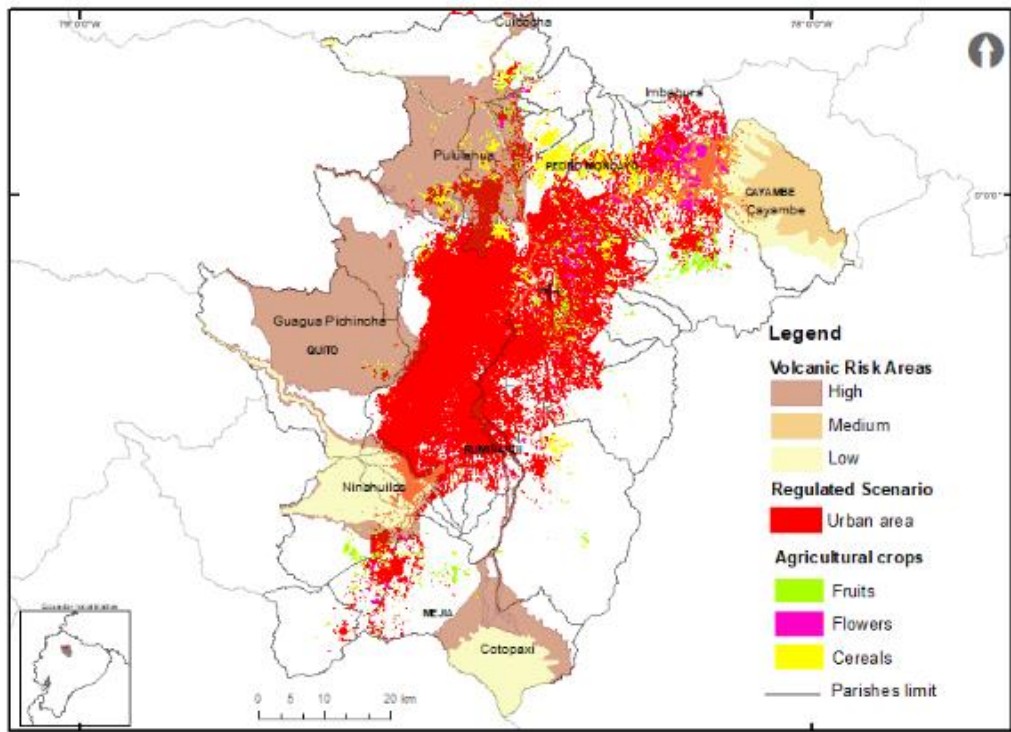

**Figure 11.** Regulated scenario—volcanic risk areas and agriculture crops, 2050. Source: Geophysical Institute of the National Polytechnic School, 2013.

The simulated land use maps of 2050 show that the trend scenario is more respectful of the agricultural and scrubland areas located in the north of Quito, in Pedro Moncayo or in Cayambe than the regulated scenario. In the regulated scenario scrub coverage disappears almost completely. There is only a small remnant of protected forest, which could produce a fragmentation of ecosystems and loss of biodiversity (the scrubland or dry forest houses unique species and is one of the ecosystems most threatened and altered by anthropogenic activities) as indicated by Ríofrio [44]. Meanwhile, the regulated scenario respects native vegetation more and restricts urban expansion towards certain natural protected areas (declared by the Ministry of Environment) such as Pululahua, but it consumes the forests that form the green belt of Quito (Figure 11). In short, the presence of natural protected areas neither prevents urban growth nor the conversion of agricultural land into urban land.

The regulated scenario promotes greater soil conversion than the trend scenario due to the IEE's urban expansion proposal which allows urban development in forests areas, volcanic risk areas and agricultural production areas (Figure 11). It is essential to review proposals for urban expansion that are consistent with natural heritage protection policies and are complemented with risk reduction policies due to volcanic threat. The new development policies starting in 2020 will discourage growth towards the periphery, reinforcing the old centralities indicated by Tapia [45]. Similarly, more coordinated planning between the different governmental entities that is not limited to regulating the territory within an administrative boundary is necessary. That is, comprehensive planning that allows achieving sustainable urban development.

The knowledge gained from these simulations confirms the need to use planning laws and instruments not just for the DMQ, but also for the planning of newly developing areas beyond the current DMQ boundary. Increased regulation and control should be applied to a wider territory, the so-called NAQM, supported by policies and spatial plans that provide a framework and instruments to manage urban sprawl.

According to Castro [46], a collaborative planning effort between the municipality of Quito and the cantons of the province of Pichincha, which together could be considered as the Quito conurbation, is anticipated. The two scenarios presented here are examples of how changes in land

use, without a well-functioning land use planning and control system, might affect Quito's progress toward greater sustainability and the need to consider scaling up the planning and management scale to the NAQM area.

## 5. Conclusions

This work has shown how the integration and evaluation of land use/cover trends and policies could be considered in a proposal for a new metropolitan area for Quito. The results obtained from the modeling of two growth scenarios through 2050, trend growth and regulated growth, foresee a substantial reduction of agricultural and natural areas as a result of urban expansion. The two scenarios allow the exploration of different results with different alternatives that can inform the debate on the desired development and planning of the territory.

Nevertheless, neither of the scenarios presents an ideal outcome. Both pay insufficient attention to nature conservation and the volcanic risk areas. Irrespective of their low legal status as development constraints, they are both important considerations for sustainability and for resilience. The lack of coordination and joint work between the governments of the cantonal municipalities and the municipality of the metropolitan area is confirmed. To this day, none of these municipalities has defined enough areas for urban expansion in their planning. As a result, urban development has long obeyed an inverse logic in which the first step is to inhabit the territory and then the later steps are to plan the city, which is why the regulations for land use planning and its instruments were created in Ecuador in 2016.

Urban planning instruments must cover the territory comprehensively. The results of this paper show that urban dynamics of Quito are not constrained by administrative limits. Planning based on a single geographical scale and space is therefore obsolete. Aspects such as globalization, natural risks, landscape quality, sustainability and urban resilience must be incorporated into the context and plan development processes.

Working with scenario modeling approaches can provide information to Latin American governments and metropolitan cities for designing desirable futures through simulations. Ex-ante modeling and evaluations support political decision-making and its spatial implications while creating opportunities to engage various agents involved in planning processes. In short, the planning of future development needs and land use conflict prevention must be inclusive, including policy makers, academics and local communities.

**Author Contributions:** Research conceptualization Esthela Salazar and Cristián Henríquez; methodology Esthela Salazar, Cristián Henríquez, Jorge Qüense and Richard Sliuzas; software analysis (Dyna-ClUE) Esthela Salazar, Cristián Henríquez, and Jorge Qüense; validation Cristián Henríquez and Richard Sliuzas; formal analysis Esthela Salazar, Cristián Henríquez and Richard Sliuzas; research analysis Esthela Salazar; resources Cristián Henríquez; data curation, Esthela Salazar, Cristián Henríquez and Jorge Qüense; writing-original draft preparation Esthela Salazar and Cristián Henríquez; writing –review and editing, Cristián Henríquez and Richard Sliuzas; visualization Esthela Salazar, Cristián Henríquez, Richard Sliuzas and Jorge Qüense; supervision Cristián Henríquez; project administration, Cristián Henríquez; funding acquisition, Cristián Henríquez. All authors have read and agreed to the published version of the manuscript.

**Funding:** This research was funded by FONDECYT/CONICYT (Chile) of grant number Nº1180268 (Projection of growth scenarios in middle cities. Spatial simulation models for urban-environmental planning) and SENESCYT (doctoral scholarship program for university teachers 2015).

**Acknowledgments:** The authors acknowledge the support received from the Doctoral Program in Geography, Pontificia Universidad Católica de Chile; and the Department of Urban Planning and Geoinformation Management, Faculty ITC of the University of Twente (Enschede, the Netherlands). Special thanks to Paulina Contreras, Lenin Henríquez-Dole, Mauricio Morales, Andrea López, Lidya Prieto and the valuable contributions of anonymous reviewers.

**Conflicts of Interest:** The authors declare no conflict of interest.

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
