# Peer review of "Evaluating Spatial Scenarios for Sustainable Development in Quito, Ecuador"

_ijgi, doi:10.3390/ijgi9030141_

Round 1

Reviewer 1 Report

he paper present application of the Dyna-CLUE model in real urban planning process for  metropolitan region of Quito. It is very comprehensive and interesting for reading. 

However, do not emphasis or developing any new methodology for urban planning but new approaches in which simulation models have become a fundamental tool for planning.

In this paper some chapters are much bigger than other which can results with to detail information in one part and insufficient information in Discussion or Conclusion chapters. Suggestion is to equalized this disbalance.

There is some technical details that need to be corrected:

After every (sub)chapter there should be some texts with explanation of the meaning of the (sub)chapter. (e.g. 2.; 2.1 and 2.1.1 or 3.2 and 3.2.1) There are some differences in text font (e.g. in row 71) Titles or sources of figures/tables should be positioned in the same page. The title of chapter 2.2.5 is the last text on the page 12 which should be connected to the chapter text. Some titles of the chapters (e.g. 3.1) are not bold like all other In some part of the presented paper authors use expression "This study presents..." (row 122) which should be corrected in "This paper presents..."

Author Response

Dear:

Reviewer 2 Report

Comments for the authors:

The manuscript presents the use of Dyna CLUE to model two scenarios of urban expansion in Quito metropolitan region and builds up a discussion on the implications for land planning. This type of research is of extreme relevance, especially in Latin America where big cities present huge trends of urban growth and conurbation. Restricting areas from urban growth is not only a challenge for urban planners, but for governments and organizations concerned with a sustainable development. Both scenarios elucidate possible futures and indicate which measures should be reinforced and what policies and drivers must be addressed to have Quito growing without affecting its own needs (with enough food provision, preventing risk from volcanos and maintaining environmental integrity with functional protected areas). This study in Quito is an example for many other Andean cities facing similar challenges, but also for several growing urban centers worldwide.

I congratulate the authors for approaching such challenging theme with up to date modeling tool and kindly ask you to take my input as a contribution to improve your manuscript.

Here I recommend some modifications to make the presentation of the manuscript more straightforward. In the attached file there are I added specific comments.

General comments

Methods can be shortened, if written in a more direct way (consider language review). I think that a spatial representation of the driving forces would be a valuable complementation to be included as supplementary material.

For the results, I strongly recommend you to present a more synthetic figure to show 1) land use in 1998, 2017 and 2050 (this last, for both scenarios) and 2) show (maybe a different figure) the land use that changed from 2017 to 2050, detached from that that has not changed. This could allow the reader to see the differences in the two scenarios more clearly.

In sections 3.2 I strongly recommend you not to split the results in three items. First, I understood that sections 3.2.1, 3.2.2 and 3.2.3 represented different scenarios. Then, I realized you were only presenting the results of the regulated scenario and emphasising different aspects in each item. I consider enough if you present these results in separate paragraphs.

I also suggest you should include a final paragraph in the Materials and Methods explaining how you evaluate the effect of the SNAP and the volcanic risk zones. It will prepare the reader for those results. Besides, I wasn’t sure the SNAP and the volcanic risk zones were used as input for the regulated scenario. This must be very clear in the scenario description.  

In section 3.2.3, and in Figure 11, it is very difficult to understand whether you are evaluating how the urban growth is projected over risk areas, or if the risk areas were considered in the scenario building, thus preventing urban growth to happen there.

This can be clarified either in the description of the scenario and by adding this analysis in Materials and Methods. Also, you can improve the explanation in section 3.2.3 as a whole. 

In the discussion, you refer to [42] and it makes me think about the marked trend of urban sprawl presented by Quito: mostly towards north and south. I believe mostly due to relief and the way the city is fixed between west and east ridges (cordilleras ocidental y oriental). I consider it is worth mentioning this trend in the introduction. I think it would add a lot to the reader if you briefly mentioned [42] and [43] in the introduction.

Because you keep the demand for land use constant in both scenarios (Table 6), it is not unexpected for me that both modeled scenarios will present similar increase in urban areas. The interesting is to evaluate how the new urban areas are spatially arranged in the landscape. And this is also a consequence of the restrictions posed by the design of the scenarios. Therefore, I ask you: besides the trend and regulated scenarios, could you propose an ideal/desired scenario considering the trends in urban population increase in Quito and its surroundings? I think you partially address the answer to this question in lines 493 to 499. I think that in this paragraph emerges your main contribution to planning the development of Quito. I think a last additional paragraph could emphasize this contribution.

The manuscript has some spelling and punctuation minor mistakes and some other language use that should be reviewed. Some references look incomplete and should be reviewed as well.

Author Response

Dear:

Round 2

Reviewer 2 Report

First of all, I thank the authors for addressing all my general comments. I am completely satisfied with the way you responded.

Unfortunately, there was a problem with the submission of my specific comments. Possibly the file was not uploaded properly, I apologize for that. I revised the new version of the manuscript and revised my specific comments. Please consider them as you judge appropriate.

Once more, I kindly ask you to take my input as a contribution to improve your manuscript.

Specific comments:

line 22 – instead of “This model helps identify” consider using “We used this model to identify”

line 47 - check spelling "special", you meant spatial?

line 52 - the way you structured the introduction is interesting. You could include some information about how you incorporated the "points of view of the main stakeholders" and briefly point out who they are.

line 57/58 - here, you are trying to convince the reader about the advantages of using scenario modelling to support decision-making. When you state "First, scenarios provide heuristic support to explain events and their consequences." I don't agree scenario analysis provide explanations; I understand they illustrate possible consequences to 'chosen' drivers. Maybe if you include a reference here, the idea can be clarified.

line 58/60 - "Spatial and statistical data alone does not make much sense, until it is related within a framework that includes the interaction of social, economic, political and technological factors." The idea you are trying to convey is not clear, consider rewriting. Also adding a reference may help.

line 87 - I don’t think the subdivisions 2.1.1 and 2.1.2 are necessary. Instead, I suggest you have 2.1 - Study area: the New Metropolitan Area of Quito (NMAQ); and then delete the subdivisions.

line 95 - consider mentioning the volcanos and their activity

line 99 - "The territory" refers to DMQ? It is clear only after checking Figure 2

line 99 – consider using “administrative zones” instead of “Zonal Administrations”

line 102 – Figure 1, consider also locating Ecuador in South America in the map; I can't read the numbers in the scale; what are Cotopaxi, Santo Domingo de los Tsachilas, Napo and Imbabura in the map? Other Provinces? Explain in the legend or remove their names. Are all the cantons of Pichincha shown on the map?

line 124 to 126 - what do you mean by “traditional zones” and “new growth fonts”?

line 130 - what type of document is this reference [15]?

line 131 - I have no idea where Cutuglaua is. Is it in Mejía?

line 134 - "of the main economic activities" of the country? of those cantons?

line 135 - " as the main export product" of the country?

line 142 - I understand the importance of the Panamericana highway, however, since these other provinces are not on the map, there is no use in listing them. I suggest you finish the phrase after "other provinces", or add them in the map (Figure 1?)

line 145 - the description you made does not lead to this understanding. Maybe you can add some other information on population growth for the other cantons to lead to this conclusion

line 145 - if I understood well, the proposition you make for the New Metropolitan Area has the intention to aggregate the 5 cantons into the territorial analysis, instead of performing them only for DMQ. Is this correct? In case it is, why do you say consolidate? was it used before? Why were not other neighbor cantons also included?

line 152 – just repeating I don’t think this subdivision is necessary, instead, I suggest you add a paragraph briefly addressing the main contents of Table 1 and maybe keep the table with less text.

line 169 - I don't think the first phrase is the right one to open the 2.2 section. The information is relevant but consider relocating it in the text.

line 171 - "whose" does not sound right for me, please check

line 177 - use the acronym Dyna-CLUE

line 179 - again "whose" doesn't sound right, please check

line 180 - by "spacing" you mean spatial?

line 190 - I suggest rewriting this part "The Dyna CLUE model was adopted to create future scenarios for the new metropolitan area of Quito based on a probabilistic method of logistic regression whose objective is to make a quantitative description of land use changes. The model’s interest is that it allows monitoring …"; to summarize and avoid repetition. Suggestion: “We simulated future scenarios for the new Quito metropolitan region using Dyna-CLUE and evaluated the quantitative outputs. This modeling allows monitoring…”

line 194 – how was the consultation included as an additional value?

line 201 - explicitly describe what "TPROPi,u" stands for

line 212 – do not call figure 4 here, it is not showing the satellite images. Refer to it when you mention the maps in line 224

line 214 – does it refer to NAMQ?  

line 221 - did you evaluate the quality of the two maps produced? (Kappa index or anything similar?)

line 231 - Table 2 and Figure 5 have the same information. Table 2 is more complete because it presents absolute and percentage values. I suggest you remove Figure 5, or keep both but remove the two % columns from table 2 to avoid information redundancy

line 236 - what do the values in the table 3 represent?

line 240 – how were the main driving forces identified? I got the information only in line 244 and 245. Consider starting the paragraph with this suggestion: “Nine main driving forces for urban growth were identified with the support of main government agencies and the literature (Table 4; Figure 6). These variables can potentially affect the specific location of land uses, for example, the main road network is a primary factor for population growth due to accessibility to the territory.”

line 242 – verb is missing: “establishments also are”

line 257 - I don't understand what the steps are for. Consider connecting this paragraph to the previous and explain what are the steps you are describing in the sequence. Instead of “These are detailed below” describe what are you explaining in the steps in the following paragraph.

line 263 – reference error (and because of this, maybe the others were not right, please check)

line 288 - what is validation 2? you didn't mention it before

line 292 – does “(1)” at the end of the line stands for “validation 1”? Please clarify.

line 302 – consider briefly explaining in the legend what the colors mean and what validation 1 and 2 are.  Also, you can remove the title embedded in the figure, it is already in the legend.

line 305 - instead of "The demands" start the phrase with "The land use demands"

line 306 - what do you mean by annually? Until here, there was no mention in the text for which years the scenarios were developed. So, if by "annually" you mean each year in the future used in the simulation, replace it by "for each year", or merge it to the explanation in the following phrase

line 308 - make it shorter, refer to the maps produced

line 313 - why is there a "*"?

line 314 - how were the land uses (1. urban, 2. moorland, 3. native vegetation, 4. without vegetation, 5. scrub, 6. agricultural, 7. bodies of water, and 8. without data) grouped into the four categories (urban, native vegetation, agriculture, scrub)?

line 321 - because of the years shown in Table 6, you should explain in the text why those years were chosen.

line 321 - legend to table 6 could be more detailed

line 329 - the first paragraph of section 2.2.5 should be moved to the introduction by merging any information provided here that is not already there. Begin the section by saying two scenarios were developed.

line 340 - clarify what you mean by "new morphology that depends on the traditional city"

line 342 - in the description of scenario 2, regulated, can you be more objective in the restrictions considered?

line 353 - consider adding a phrase at the end of this paragraph to lead the reader to the following subtopics (results of both scenarios).

line 361 - consider adding a map with main roads as a supplementary material

line 362 – most people will not know what or where Papallacta is. You could clarify or remove it.

line 364 - I understand, from Table 7 that 23% is the percentage area of the NQMA with urban land use. This represents an increase of 10% in 33 years, it that right? If this correct, you must rewrite the way you present the percentages.

line 366 – suggestion: add a plane icon in your maps, so that the readers know the location of the new airport

line 366 - do you mean northeast instead of "northwest"?

line 368 - what do you mean by "urban spot"?

line 369 - what do you mean by "the greater metropolitan city"? Is it a reference to Quito main urban area?

line 370 - Machachi was never mentioned before and it is not in your maps, so I have no idea where it is

line 370 – to which figure do you refer as Figure 8a? If you mean figure 9a, there is no reference to figure 8 until here. Please revise the figures numbering.

line 372 - in Table 7 you present the amounts of each land use for the trend scenario. Is there any chance mapping procedures of 1998 and 2017 satellite images to be the cause of increase in natural vegetation? Does it represent actual increase?

line 374 - instead of "reduced to 17%", use " reduced from 21% to 17% in 33 years"

line 377 - instead of "up to 53%", use "from 58% to 53%"

line 378 to 380 – I really appreciate your description; however, it would be better understood if you showed these places in a map

line 386 - this paragraph is repeating the method to establish the scenario. Either delete or relocate the information. I think this Figure 8 should be presented in the methods when you describe scenario 2.

line 389 - what do you mean here? "However this information is not binding for the planning, it is only indicative"

line 393 - consider mentioning it is only 2% larger

line 397 - it would be very useful to have the Ilaló volcano located in the map

line 398 - consider saying that the loss is only 2% greater

line 402 – two-thirds of the information in table 8 was already presented in table 7. Consider merging both tables in one.

line 406/407 - here you mention specific names of locations not shown on the map. I suggest you add some references to the map.

line 418 – the same information in figure 9b is presented in tables 7 and 8. I think that only a table merging tables 7 and 8 is enough to present this result. Figure 9b is not necessary.

line 420 - what is the result presented in figure 10? You must describe in Material and Methods this analysis. This is a scenario or the analysis of the overlap of agricultural areas and urban growth predicted in the regulating scenario? The representation in Figure 10, the way it is, does not show where urbanization took place over agricultural areas, for this, a map algebra could show the urban areas (in 2050) that used to be agriculture (in 2017).

line 433 - could you say unexpected instead of "peculiar"?

line 433 – what do you mean by "naturally"?

line 434 to 437 - the sentence is too long; I suggest breaking the ideas in shorter sentences

line 447 - with "becoming" do you mean that because of their location they are a "main source of food supply"? If so, I suggest you rewrite it.

line 448 - replace "as revealers of" by revealing

line 449 - consider removing "mutations and"

line 487 - in both scenarios?

line 488 – I suggest keeping this phrase in the previous paragraph, not separated in another paragraph.

line 516 – urban resilience? Society’s resilience? Whose resilience?

Author Response

Dears, the time was very short (5 days) we worked very hard. We hope living up to expectations.

Regards,

Esthela, Cristian, Jorge and Richar.
